# The Workflow of a New Dynamic Navigation System for the Insertion of Dental Implants in the Rehabilitation of Edentulous Jaws: Report of Two Cases

**DOI:** 10.3390/jcm9020421

**Published:** 2020-02-04

**Authors:** Armando Lopes, Miguel de Araújo Nobre, Diogo Santos

**Affiliations:** 1Implantology Department, Maló Clinic, 1600-042 Lisbon, Portugal; alopes@maloclinics.com (A.L.); dsantos@maloclinics.com (D.S.); 2Research and Development Department, Maló Clinic, 1600-042 Lisbon, Portugal

**Keywords:** dental implants, digital, all-on-4, zygomatic implants, navigation. zygoma, maxilla, navigated surgery

## Abstract

Background: This case series describes the surgical workflow during maxillary full-arch rehabilitations in two patients through the All-on-4 concept (standard and hybrid) assisted by DTX Studio Implant Software planning and X-Guide Navigation. Results: The X-Guide Navigation enabled the drills and implants to be positioned and oriented precisely, allowing the implants to be positioned favorably under the surgical and prosthetic points of view through the avoidance of damage to the maxillary sinus and nasal cavity. Dynamic navigation-assisted surgery provided advantages, including the possibility to modify the implants’ system, length, or location perioperatively. However, it must be underlined that to achieve proficiency with this technology it is necessary to consider a necessary learning curve. Conclusion: The insertion of dental implants assisted by dynamic navigation for maxillary full-arch rehabilitations was considered a safe and predictable procedure. Nevertheless, it can be improved (such as with a simpler fiducial markers protocol), aiming to simplify the procedure.

## 1. Introduction

The All-on-4 treatment concept consists of an immediate function rehabilitation protocol for full-arch rehabilitation of edentulous arches with insertion of four implants in the anterior region (two axial implants inserted anteriorly and two distally tilted posterior implants) to support a fixed prosthesis [1,2]. This concept was further developed to include the use of zygomatic implants alone or in combination with standard implants in situations of extreme maxillary resorption (All-on-4 Extra-maxilla or All-on-4 Hybrid) [3,4].

The rehabilitation of atrophic edentulous jaws is a challenge, considering the limited bone quality and quantity, providing an additional difficulty for oral surgeons to place implants in areas that favor the probability of success. New dynamic navigation systems allow the dental surgeon to locate the continuous updated instruments position during surgery and provide accurate spacing and angulation of the implants [5,6,7]. Compared with free-hand modality, the placement of dental implants assisted by navigation and virtual planning enables the implants’ accurate angulation and spacing. Moreover, virtual implant planning and navigation provide surgical and prosthetic collaboration for achieving a successful outcome through exact planning and the precise arrangement of the plan. Nevertheless, it is necessary that more research is conducted to provide an outcome evaluation of assisted dynamic navigation implant-supported surgical procedures. The aim of this clinical case series was to describe the workflow of full-arch maxillary rehabilitations through the All-on-4 concept (Nobel Biocare AB, Gothemburg, Sweden) using only standard implants or a combination of standard and zygomatic implants assisted by dynamic navigation.

## 2. Case Reports

The present case series illustrates the surgical workflow during maxillary full-arch rehabilitations through the All-on-4 concept assisted by dynamic navigation (using the digital workflow protocol previously described) [6] in a standard All-on-4 maxillary rehabilitation (Nobel Biocare AB) [1] and in a zygomatic implant insertion in an All-on-4 Hybrid rehabilitation (Nobel Biocare AB) [3]. This research was carried out in accordance with the Declaration of Helsinki. Both patients provided written informed consent for participating.

### 2.1. Patient 1

A 57-year-old female patient presented with the need for maxillary full-arch rehabilitation. The patient’s main concern was aesthetics and function. A review of the patients’ medical chart was performed and clinical examinations were supplemented by an orthopantomography (Figure 1 and Figure 2) and cone beam computerized tomography (CBCT) scan and intra-oral scanner (Trios, 3 shape A/S, Copenhagen, Denmark) were performed.

Considering the surgical protocol, the intervention was performed using local anesthesia with articaine chlorhydrate with epinephrine 1:100,000 (Scandinibsa 2%, Inibsa Laboratory, Barcelona, Spain) [1]. Prior to surgery, the patients were administered diazepam (Valium 10 mg, Roche, Amadora, Portugal). Antibiotic medications (amoxicillin 875 mg + clavulanic acid 125 mg, Labesfal, Campo de Besteiros, Portugal) were given 1 h before surgery and daily for 6 days thereafter. Corticosteroids (prednisone (Meticorten Schering-Plough Farma Lda, Agualva-Cacém, Portugal), 5 mg) were administered daily in a regression mode (15 to 5 mg) between the day of surgery and 4 days post-operatively. Anti-inflammatories (ibuprofen, 600 mg, Ratiopharm Lda, Carnaxide, Portugal) were given between day 4 and 7 post-operatively. Analgesic medication (clonixine (Clonix, Janssen-Cilag Farmaceutica Lda, Barcarena, Portugal), 300 mg) was given on the day of surgery and only if needed on the first 3 days post-operatively. Antacids (Omeprazole, 20 mg, Lisbon, Portugal) were administered between the day of surgery and the day 6 post-operatively [1].

A mucoperiosteal flap was raised along the top of the ridge with relieving incisions on the buccal aspect in the molar area. Roots and teeth were extracted and extraction sockets were cleaned by means of curettage. Bone crest regularization was performed, aiming to achieve a stable and leveled platform (Figure 3).

Five TriStar screws (Impladent, LTD, Jamaica, NY, USA) were attached in the maxilla (Figure 4 and Figure 5), working as edentulous fiducial markers, and a CBCT scan was performed.

The DTX Studio Implant Software (Nobel Biocare, Zurich, Switzerland) was used to evaluate the baseline clinical situation and study the best surgical and prosthetic solution. Considering the patient’s local site conditions, the position and size of the implants were selected in the DTX Studio Implant Software (Figure 6).

After the CBCT scan with TriStar screws, the X-Guide clip receptor was fixated in the maxilla and the navigation surgery was initiated (Figure 7 and Figure 8).

The surgical staff followed the prompts in the X-Guide software (X-Nav Technologies, LLC, Lansdale, PA, USA). Instruments and edentulous fiducial markers calibration (handpiece tracker, chuck, probe tool, 5 TriStar, and the preparation drills) (Figure 9) was as follows: the overhead blue lights were reflected by the arrays of the clip receptor, which contained fiducial markers, and read by two cameras, providing the instruments’ tridimensional position and displayed in the navigation system. The handpiece also holds an array that, when combined with the fiducial markers on the clip receptor, provided accurate navigation through triangulation. Care had to be taken to ensure that the remaining instruments were in the cameras’ field of vision to be precisely tracked on the monitor.

The preparation of the implant osteotomy was made using the precision drill, 2.0 mm, 2.4–2.8 mm, and 3.2–3.6 mm drills (Nobel Biocare AB), always considering the spatial position according to the live navigation data (Figure 10 and Figure 11).

The X-Guide system (X-Nav Technologies) tracked the patient and surgical instruments and presented the real-time position and guidance feedback on a computer display for every step of the surgery (Figure 12 and Figure 13). Thus, the accurate and precise implant placement, in terms of depth (14.4 mm) and angulation (0.7 degrees), led to the most favorable surgical and prosthetic positions (Figure 13). This way, it was possible to avoid important anatomic structures, such as the maxillary sinus and nasal cavities. Four NobelParallel CC implants (Nobel Biocare AB) were placed following the All-on-4 concept: 4.3 mm × 15 mm implants placed posteriorly and 3.75 mm × 15 mm implants placed anteriorly and all implants had insertion torques greater than 35 N/cm.

The implants’ position was located between the anterior wall of the maxillary sinus, reaching an angulation of 30 to 45 degrees in relation to the occlusal plane. The emergence of the posterior implant was positioned at the first molar position (bilaterally) due to inserting the implant along the anterior sinus wall with distal tilting. The two anterior implants were inserted in an axial position and emerged bilaterally at the lateral incisor position (Figure 14).

Multi-unit plus abutments (Nobel Biocare) were attached to the implants under the following conditions: 30° 4.5 mm of height on the posterior implants and straight 3.5 mm and 2.5 mm abutments for the anterior implants (implant **#**12 and **#**22, respectively) (Figure 15).

Non-resorbable sutures (3-0, B Braun Silkam, Aesculap Inc., Center Valley, PA, USA) were used to close the flap. The implants’ insertion with a primary stability of 35 N/cm allowed for connecting a fixed prosthesis on the same day of surgery, achieving immediate function: a provisional pre-made, high-density, acrylic-resin prosthesis (PalaXpress, Kulzer Hanau, Germany) with 12 teeth (Mondial and Premium teeth, Kulzer) was connected on the day of surgery (Figure 16, Figure 17 and Figure 18).

### 2.2. Patient 2

A hypertensive female patient of 70 years of age presented with the need for completion of the maxillary full-arch rehabilitation after one previously inserted tilted implant failure. The patient’s medical chart, clinical examinations, and orthopantomography were updated (Figure 19 and Figure 20) and a CBCT scan was performed.

For the posterior implant, considering the maxillary bone quantity classified as a D-V Cawood and Howell classification [8] and the impossibility of placing a conventional implant on the residual bone, one implant with zygoma anchorage was planned. However, instead of using the conventional total edentulous workflow for dynamic navigation described in the previous case (screwing the TriStar screws and fixating the X-guide receptor), the protocol was modified. Using a conventional navigation clip for single and partial dentate cases and because there were no remaining teeth in the maxilla, the two anterior straight implants were used to fixate the clip. A silicone impression was made of the maxillary implants and, in the dental laboratory, the clip was relined to the two anterior implants on top of the stone model, becoming screw-retained (modified clip, Figure 21). With this change, the modified clip was used during the CBCT and all of the surgical protocol was simplified as the modified clip could be stabilized to the maxilla, avoiding the need of screwing the five TriStar screws and the X-guide receptor into the bone.

The DTX Studio Implant Software (Nobel Biocare, Zurich, Switzerland) was used to assess the best surgical position for the insertion of a zygomatic implant on the 3rd sextant through the extra-maxillary surgical technique for completion of the planned All-on-4 Hybrid rehabilitation (three standard implants and one zygomatic implant).

Considering the surgical protocol [3], the surgery was performed under local anesthesia. A mucoperiosteal incision was performed along the crest of the ridge, slightly palatal, from molar to canine area, with two vertical-releasing incisions over the zygomatic process (Figure 22).

Instruments and the modified clip were calibrated, following the previous description detailed in Figure 9, and the navigation surgery was initiated. The osteotomy channel begun as posterior as possible at maxillary crest level with a channel drill that defined the implant direction, keeping an approximately 3 mm safe distance from the posterior-inferior edge of the zygomatic bone, making sure that the membrane was not damaged. The sinus membrane was carefully pushed away from the external wall of the sinus, making sure that it was not damaged; this channel allowed access to the Zygoma bone with the implant drills without any tissue interference.

A round bur and a 2.9 mm twist zygoma drills (Nobel Biocare AB) were used to perform and define the extra-maxilla zygoma implant osteotomy (Figure 23 and Figure 24).

The subsequent osteotomy was performed using the subsequent drills (3.5, 4.0, and 4.4 mm twist drills, Nobel Bicoare AB) considering the spatial position provided by the system using live navigation data (Figure 25 and Figure 26).

The X-Guide (X-Nav Technologies) enabled tracking both patient and surgical instruments, featuring real-time position and guidance visualized on the computer display (Figure 27 and Figure 28). This enabled the insertion of the zygomatic implant with accuracy and precision while avoiding damaging important anatomic structures, such as the infra-orbital nerve and the base of the orbit. One NobelZygoma 0° 45 mm implant (Nobel Biocare) was inserted through the extra-maxillary surgical technique with an insertion torque over 50 N/cm, following the All-on-4 Hybrid concept.

The zygomatic implant emerged at the second premolar position at the crest level benefiting from the extra-maxillary surgical technique with zygomatic bone anchorage and only accommodated in the maxilla (no maxillary anchorage) (Figure 29).

A multi-unit abutment of 45° 6 mm of height (Nobel Biocare) was connected to the zygomatic implant (Figure 30).

Non-resorbable sutures (3-0, B Braun Silkam, Aesculap Inc., Center Valley, PA, USA) were used to close the flap. The implant insertion with a primary stability over 50 N/cm allowed for immediate function with the connection of the fixed metal-acrylic prosthesis with titanium infrastructure and high-density acrylic-resin (PalaXpress, Kulzer Hanau, Germany) with 12 teeth (Mondial and Premium teeth, Kulzer) (Figure 31) re-adapted to the position of the zygomatic implant.

## 3. Discussion

The predictability of the rehabilitation process described in the present case reports provided the necessary stability to connect an implant-supported fixed prosthesis that enabled comfort, aesthetics, and function to the patient through the implants’ functional osseointegration period. Various advantages arise from the use of guidance methods for implant placement, including the accuracy of implant placement, cost-effectiveness, and less chair-time [9].

A previous review compared the use of dynamic navigation with static guides during implant placement: [5] The main differences between both modalities concern the use of computed tomography-generated data to create computer-aided design (CAD) and computer-aided manufactured (CAM) static guides (stents) used during surgery to place implants in a predetermined position without the possibility of changing the implant position perioperatively, while for the dynamic navigation system, the operator uses a triangulation setup provided by the computer to guide the implant placement. Potential advantages and disadvantages arise from both modalities. The static guides provide the possibility of a less invasive surgery through a flapless approach without compromising the accuracy of implant placement while enabling the manufacturing of the fixed implant-supported prosthesis preoperatively [9] and less pain perceived by the patient compared to flap surgery [10]. Nevertheless, there are important limitations, including the presence of teeth that interfere with the planning of implant placement, insufficient bone volume or insufficient mouth opening (less than 50 mm) to accommodate the surgical instruments, the impossibility of changing the implant system or type, and the inability to manipulate the soft tissues in the presence of inadequate width of keratinized mucosa around the implant [5,9,11]. All these factors may pose contraindications for implant placement using static guidance, challenges that in a large proportion can be potentially overcome by the dynamic navigation system. There are several advantages related to the use of dynamic navigation-assisted surgery: first, its accuracy is accomplished independently of CAD-CAM stents that, when fabricated with any inaccuracy, can introduce significant perioperative complications [12]; second, the dynamic navigation system provides time- and cost-effective surgery by enabling the patients to be scanned and surgery planned on the same day as surgery [5,7] to perform immediate function and to manufacture the implant-supported fixed prosthesis in reduced time; third, the possibility of performing treatment independently of the patients’ limited mouth opening capability [13]; and fourth, the freedom to change the implant size, system, and location during the surgical procedure, given the possibility of a direct view of the surgical field [14,15].

Despite the comparable outcomes between static and dynamic navigation guidance in implant placement, a potential difference between the two treatment modalities was highlighted in a recent in vitro study: although a non-significant difference in accuracy at both the coronal and apical evaluations was registered, a significant difference was observed in angular deviations favoring the static guidance [15]. This may be related to the difficulty in keeping sight of the display during the surgical procedure (in the sense of being counter-intuitive for the surgeon as noted in the present case reports), a situation clearly related to the requirement of a learning curve. This finding was previously reported in a study aiming to compare the accuracies between the navigated system, the laboratory guide, and freehand drilling in cast models: The authors registered a higher accuracy in total errors at the entry and apex, lateral error and apex, and angular error for the navigated system and laboratory guide compared with freehand drilling [14]. Additionally, despite the greater attention needed to operate, the navigation system demonstrated higher accuracy in angular error compared to the laboratory guide, a situation that may be related with the necessary sleeve tolerance in the laboratory guide (the space between the metal ring and the sleeve and between the sleeve and the drill to allow cooling water circulation and drill rotation) [14]. Moreover, a learning curve is also required for the team involved in the surgical procedure in order to perform a rapid and predictable implant placement. In the management of complete edentulous cases, as in the case reports depicted in this manuscript, there is room for improvement: a simpler fiducial markers protocol could make the surgery faster and less complicated given that the TriStar and receptor fixation are time consuming and could limit the implants’ position.

Given the limitations of the present case series, the authors suggest future clinical studies be performed to investigate the outcome of the dynamic navigation system concerning angular differences when compared to freehand treatment modalities and potential implications for perioperative complications and the implant survival outcome.

## 4. Conclusions

Within the limitations of this case series, the full-arch maxillary rehabilitation through the insertion of dental implants assisted by dynamic navigation was safe and predictable. Thus, the implants were precisely inserted considering the crestal position, angulation, and depth. The system could be improved in order to provide a simpler management of full-arch rehabilitations.

## Figures and Tables

**Figure 1 jcm-09-00421-f001:**
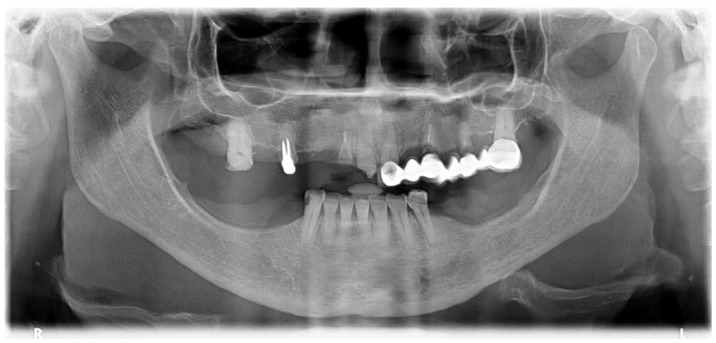
Preoperative orthopantomography of the maxilla.

**Figure 2 jcm-09-00421-f002:**
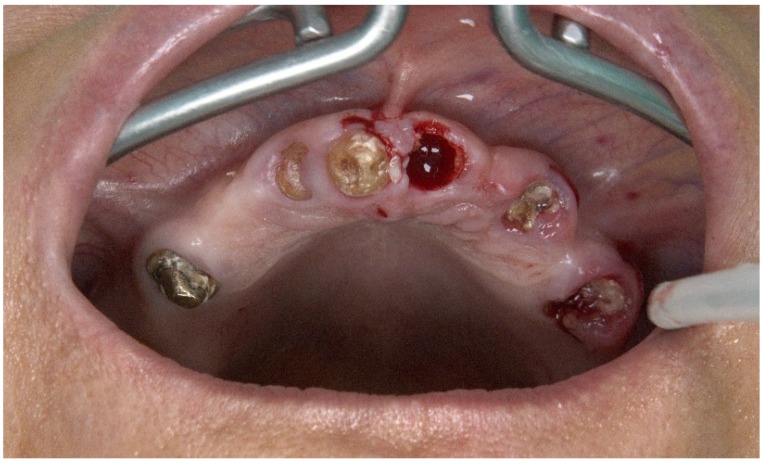
Preoperative intraoral photograph exhibiting the occlusal aspect of the maxilla.

**Figure 3 jcm-09-00421-f003:**
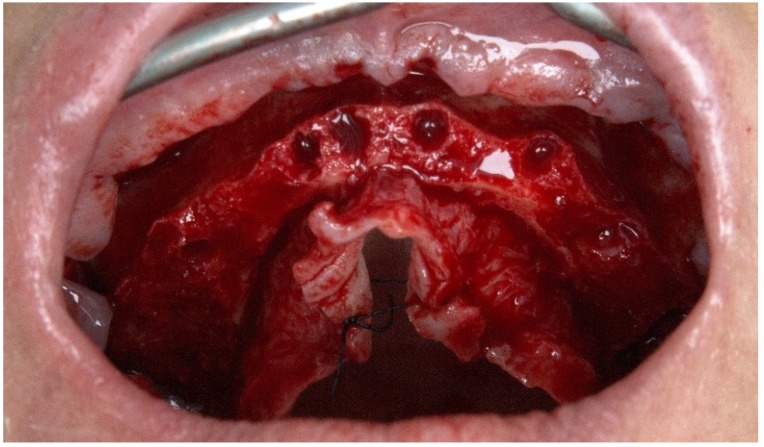
Perioperative photograph after flap opening with teeth extraction and bone regularization.

**Figure 4 jcm-09-00421-f004:**
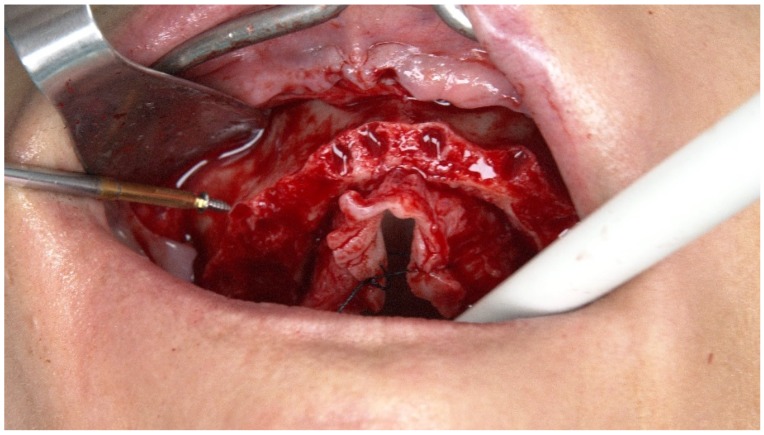
Perioperative intraoral photograph illustrating the insertion of the TriStar screws.

**Figure 5 jcm-09-00421-f005:**
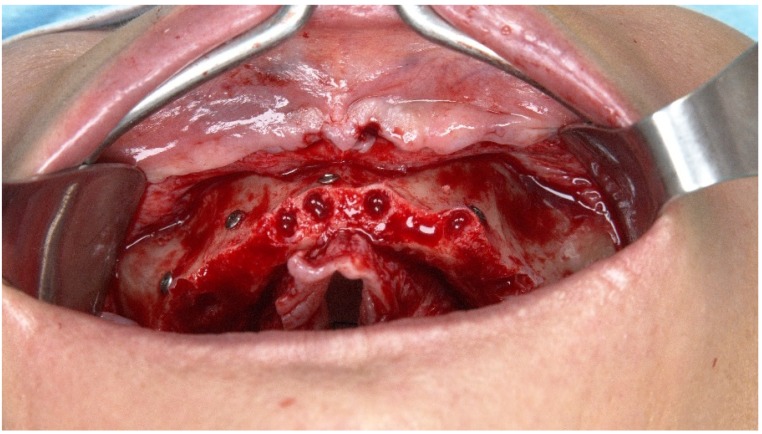
Perioperative intraoral photograph after the TriStar screws insertion.

**Figure 6 jcm-09-00421-f006:**
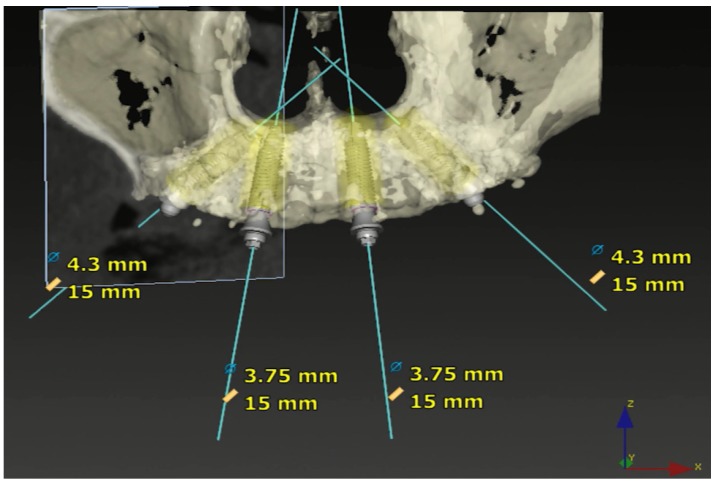
DTX Studio Implant Software (Nobel Biocare) image after TriStar screw fixation exhibiting the implants’ position, diameter, and length.

**Figure 7 jcm-09-00421-f007:**
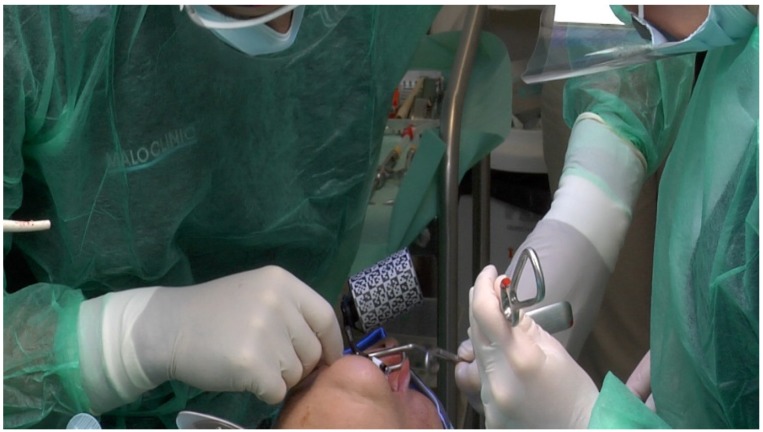
Perioperative photograph illustrating the fixation of the X-Guide clip receptor and array.

**Figure 8 jcm-09-00421-f008:**
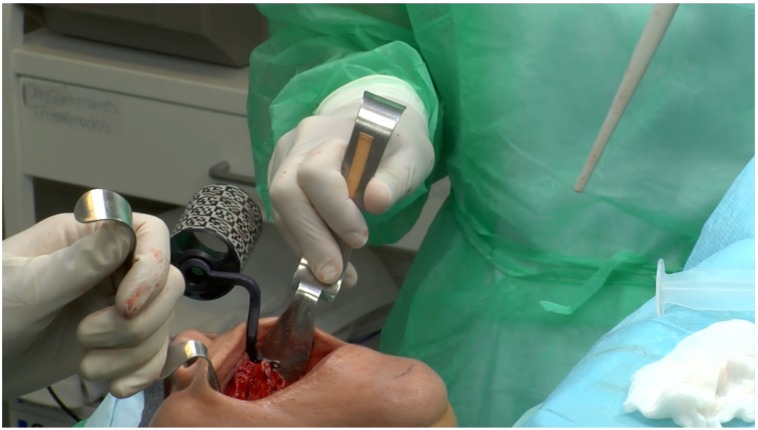
X-Guide clip receptor in place on the patient’s mouth and attached to the extra-oral array.

**Figure 9 jcm-09-00421-f009:**
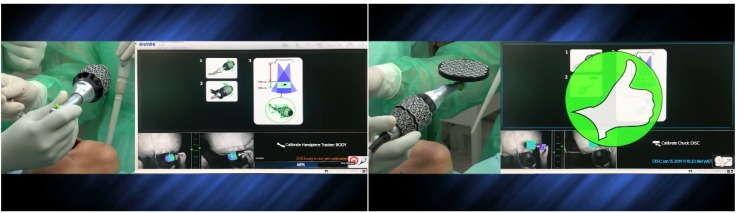
Calibration of the handpiece. The same procedure was performed for the chuck, probe tool, 5 TriStar screws, and the preparation drills.

**Figure 10 jcm-09-00421-f010:**
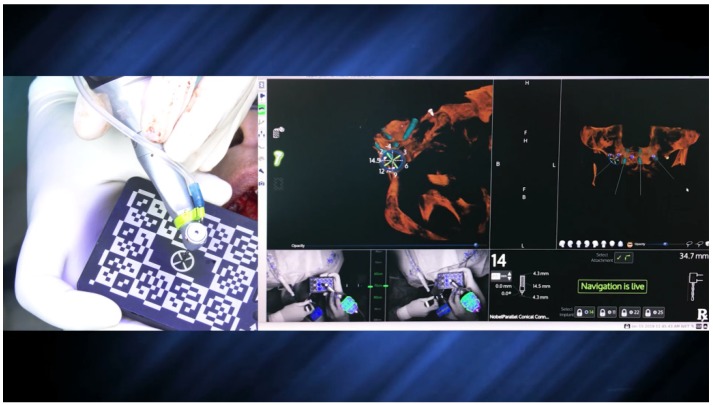
Spatial calibration of the 2.0 mm drill using an array immediately before the preparation.

**Figure 11 jcm-09-00421-f011:**
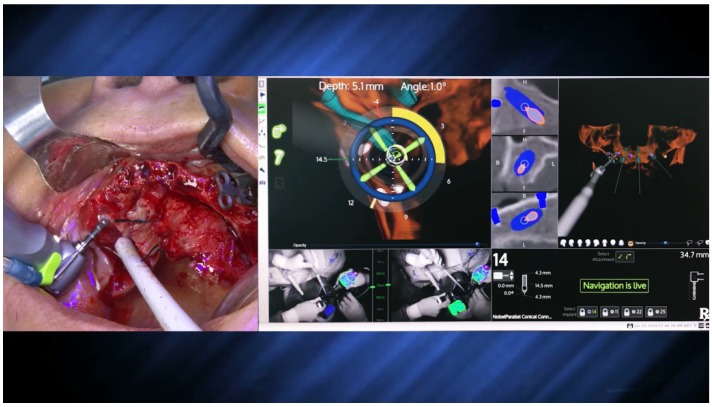
Perioperative preparation of the posterior implant (placed in a tilted position) with the 2.0 mm drill using live navigation. Note the data on the right-hand side considering depth and angle.

**Figure 12 jcm-09-00421-f012:**
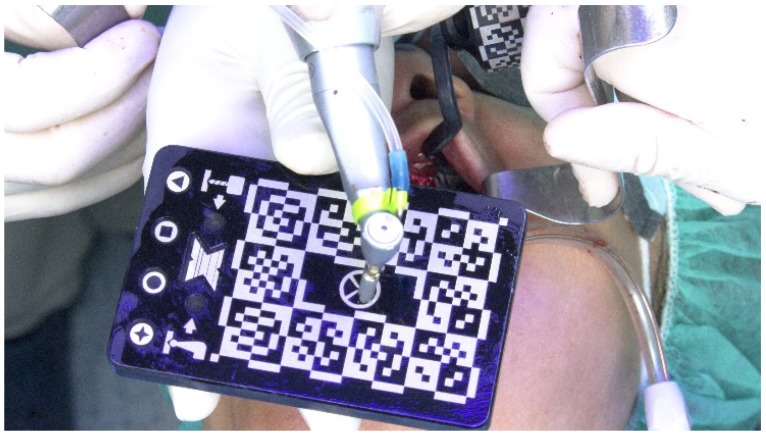
Spatial calibration of the dental implant using an array immediately before insertion.

**Figure 13 jcm-09-00421-f013:**
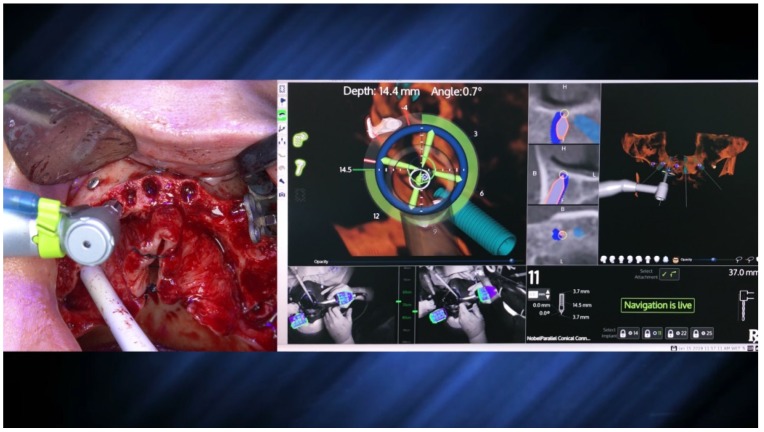
Perioperative and live navigation photograph of the implant insertion. Note the data on the right-hand side demonstrating the precision of implant insertion provided by the system, considering depth (14.4 mm) and angle of the implant (0.7 degrees), towards the target of insertion retrieved by the system.

**Figure 14 jcm-09-00421-f014:**
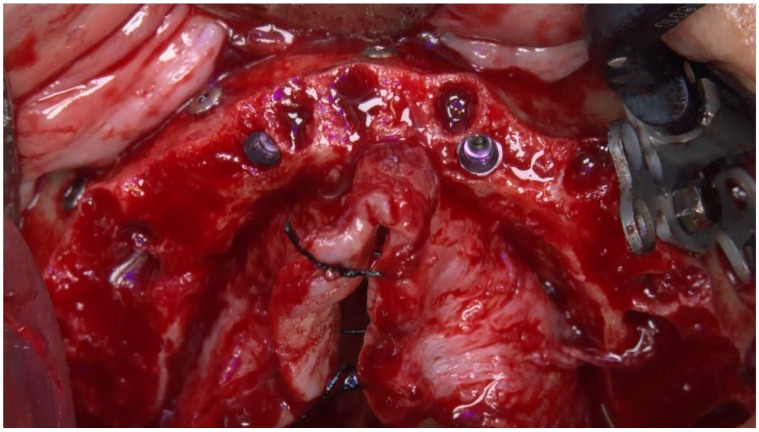
Perioperative photograph immediately after insertion of the implants using an All-on-4 concept design for rehabilitation of the edentulous maxilla.

**Figure 15 jcm-09-00421-f015:**
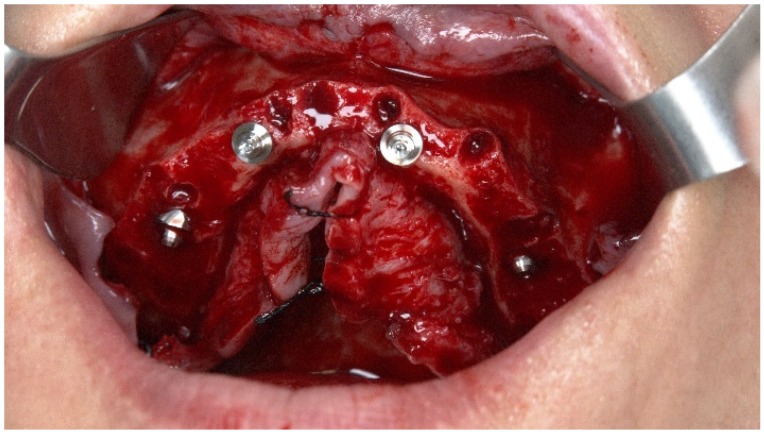
Perioperative photograph after connection of the abutments.

**Figure 16 jcm-09-00421-f016:**
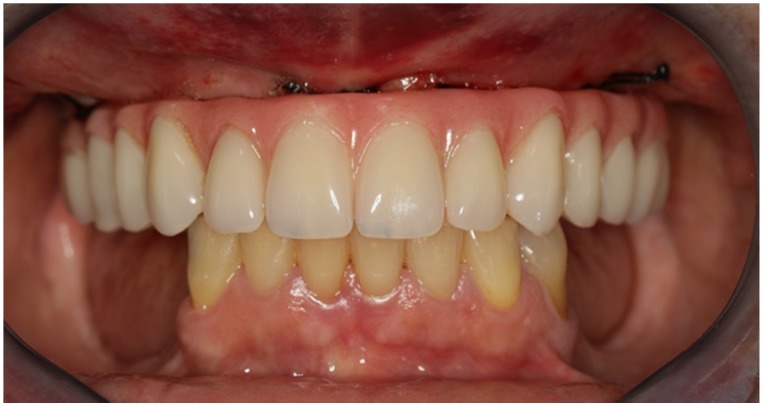
Intraoral frontal view perioperative photograph with the immediate prosthesis connected.

**Figure 17 jcm-09-00421-f017:**
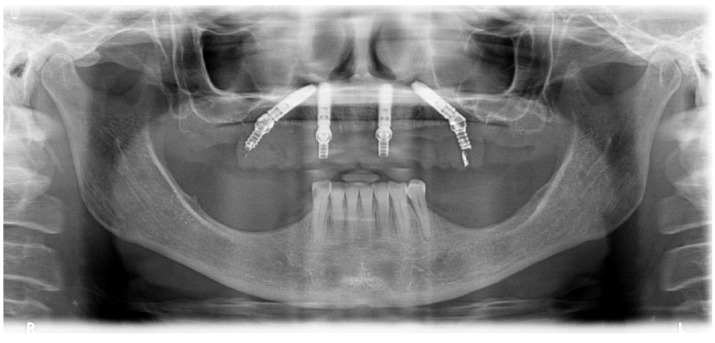
Post-operative orthopantomography of the maxillary rehabilitation through the All-on-4 concept using live navigation.

**Figure 18 jcm-09-00421-f018:**
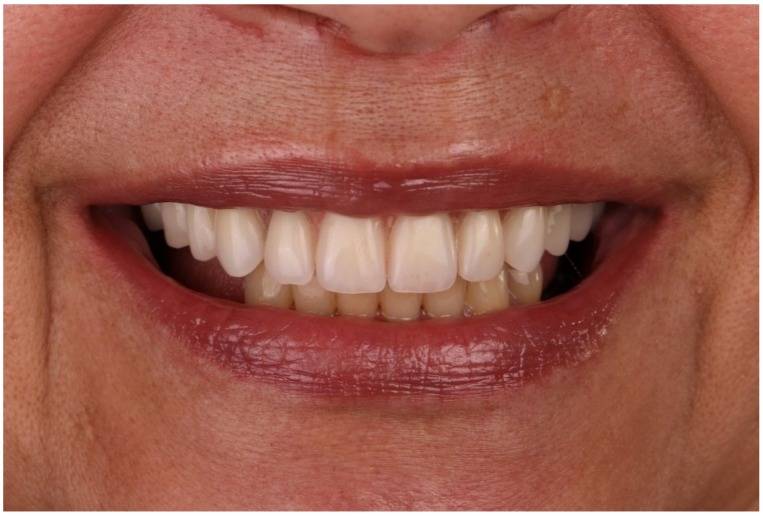
Patient smiling with the immediate full-arch implant-supported fixed prosthesis after 6 months of follow-up.

**Figure 19 jcm-09-00421-f019:**
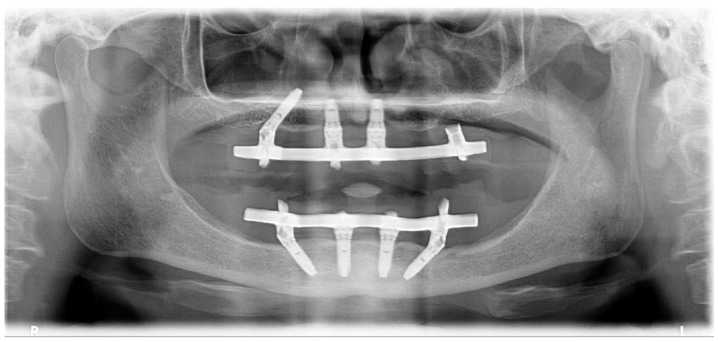
Preoperative orthopantomography of the maxilla missing the posterior tilted implant on the 3rd sextant.

**Figure 20 jcm-09-00421-f020:**
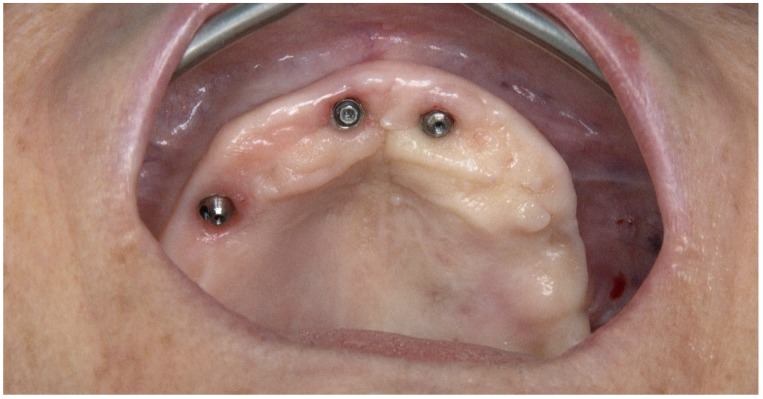
Pre-operative intraoral photograph exhibiting the occlusal aspect of the maxilla missing the posterior tilted implant on the 3rd sextant.

**Figure 21 jcm-09-00421-f021:**
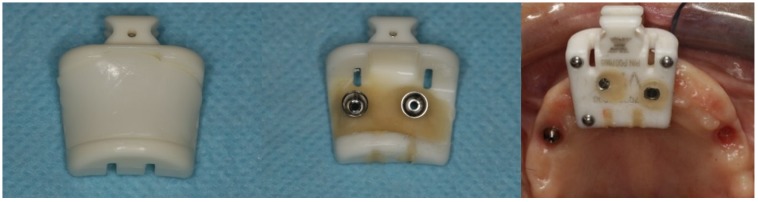
Perioperative photograph of the modified clip adapted from a conventional navigation clip for single and partial dentate cases.

**Figure 22 jcm-09-00421-f022:**
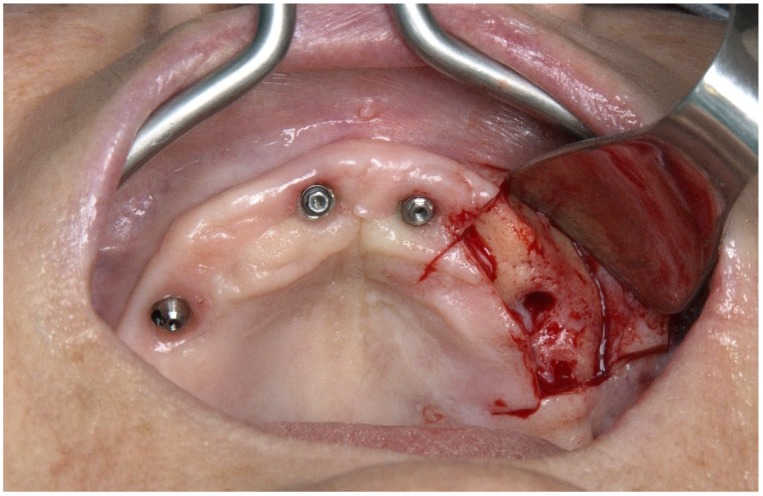
Perioperative photograph after flap opening.

**Figure 23 jcm-09-00421-f023:**
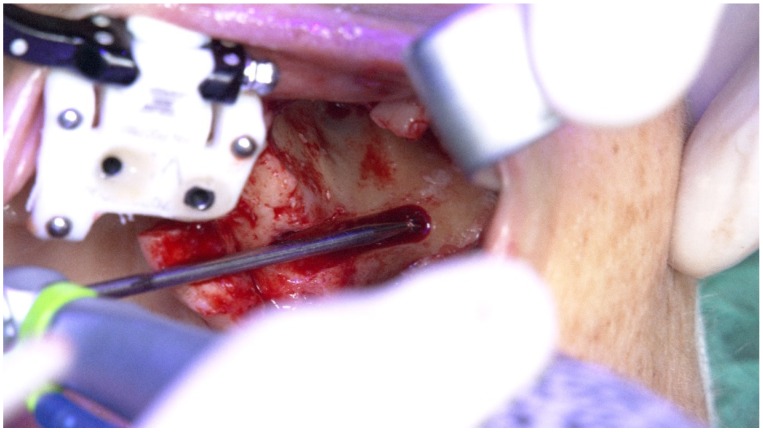
Perioperative photograph illustrating the use of the round bur for zygoma implant preparation.

**Figure 24 jcm-09-00421-f024:**
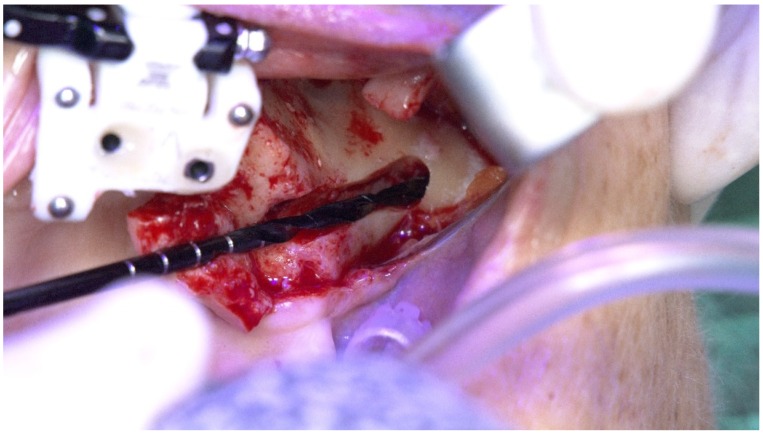
Perioperative photograph illustrating the use of the 2.9 mm twist drill for zygoma implant preparation.

**Figure 25 jcm-09-00421-f025:**
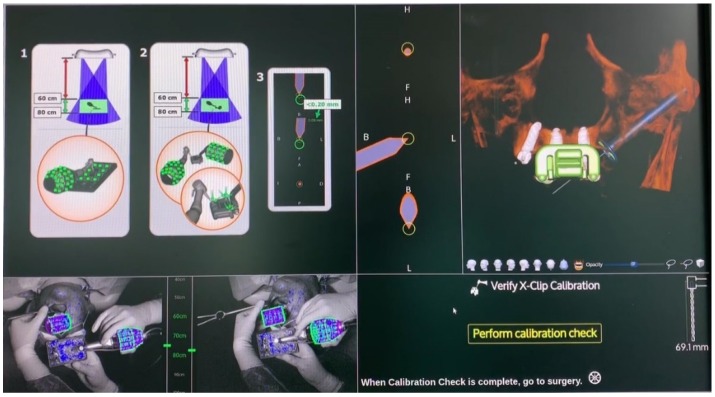
Spatial calibration of the drill immediately before the preparation.

**Figure 26 jcm-09-00421-f026:**
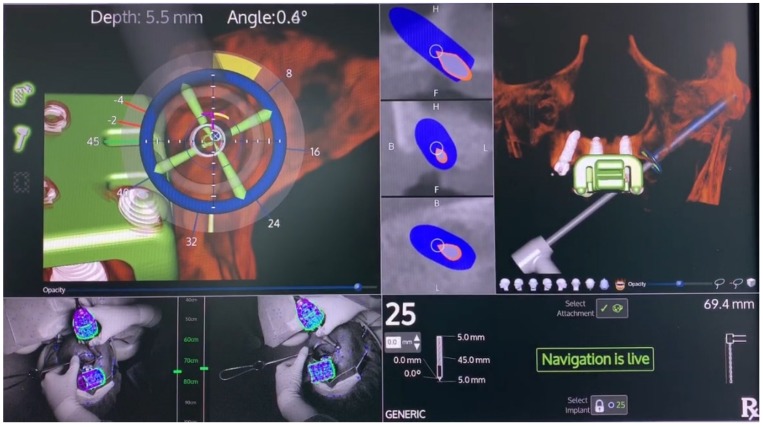
Perioperative preparation of the zygomatic implant using live navigation. Note the data on the image considering depth and angle.

**Figure 27 jcm-09-00421-f027:**
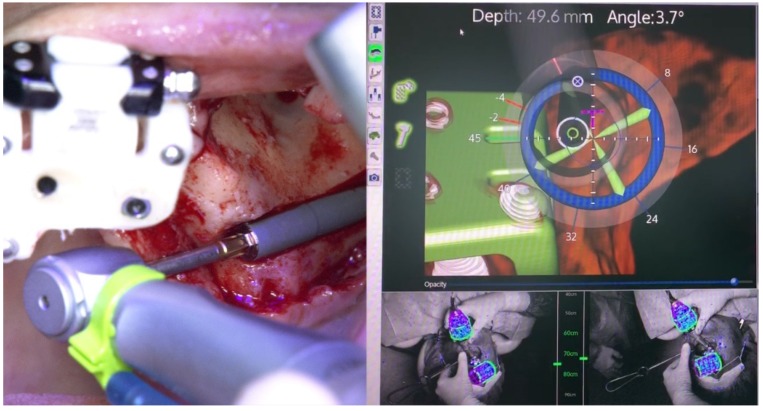
Perioperative and live navigation photograph of the zygomatic implant insertion. Note the data on the right-hand side demonstrating the precision of implant insertion considering depth (49.6 mm) and angle of the implant (3.7 degrees) towards the target of insertion retrieved by the system.

**Figure 28 jcm-09-00421-f028:**
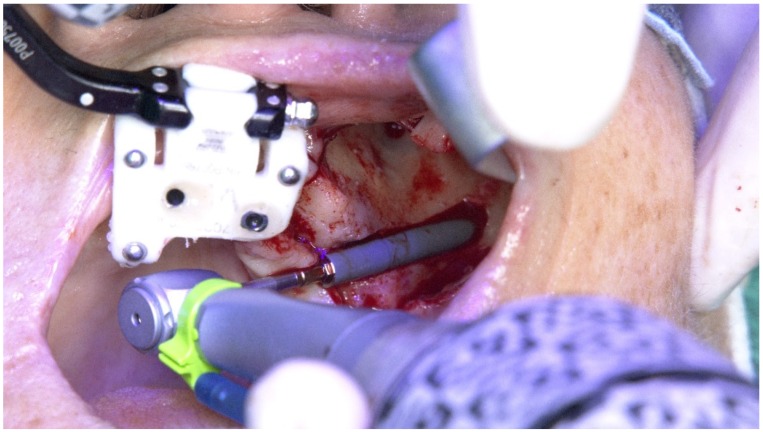
Perioperative photograph during the zygomatic implant insertion.

**Figure 29 jcm-09-00421-f029:**
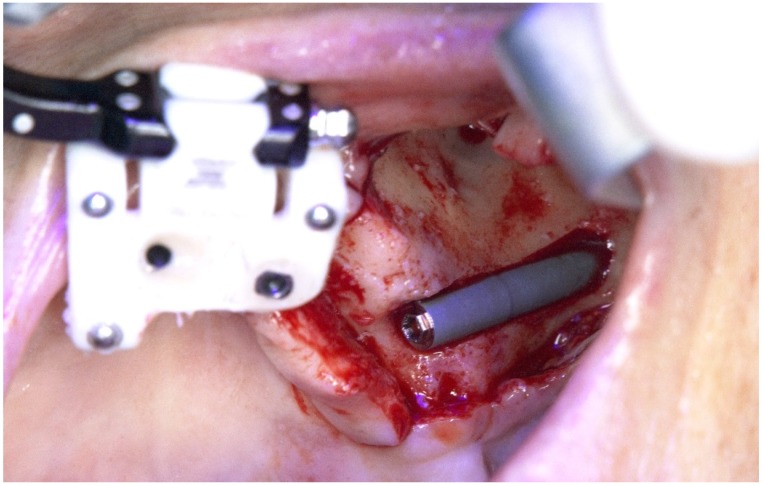
Perioperative photograph immediately after insertion of the zygomatic implant for completion of the All-on-4 Hybrid concept design for the rehabilitation of the edentulous maxilla.

**Figure 30 jcm-09-00421-f030:**
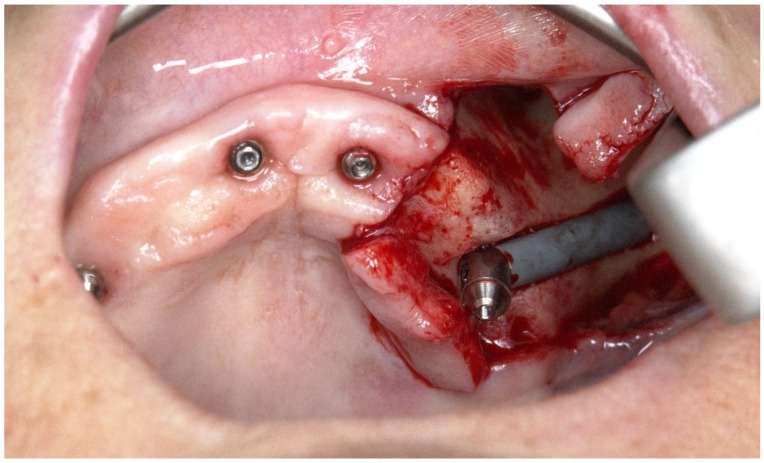
Perioperative photograph after connection of the 45° abutment.

**Figure 31 jcm-09-00421-f031:**
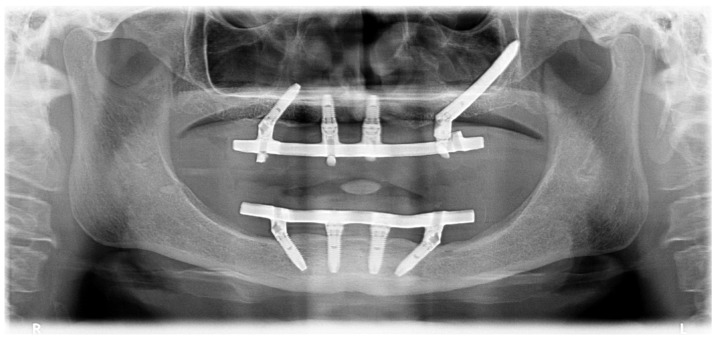
Orthopantomography with the definitive prosthesis connected one year after the intervention.

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
