# Peer review of "The Workflow of a New Dynamic Navigation System for the Insertion of Dental Implants in the Rehabilitation of Edentulous Jaws: Report of Two Cases"

_jcm, 2020, doi:10.3390/jcm9020421_

Round 1

Reviewer 1 Report

Thank you for the privilege of reviewing your work, “The workflow of a new dynamic navigation system for the insertion of dental implants in the rehabilitation of edentulous jaws: Report of two cases.” 

     This paper deals with one of the important issues in dentistry.  However, the content of this paper seems to be more suitable for Case Report rather than Research article since  (1) the workflow and the use of X-Guide had been previously described (Line 51), and (2) the result section of the abstract apparently lack of any experimental results.  Therefore, it is recommended to rewrite the paper in the structure of a case report. 

Other comments for minor revisions are the followings,

1. Line 13, The background consisted of 3 lines in one sentence, which is not easy to read. Please consider to rewrite it.

2. Line 25, “it could benefit from improvements in order to simplify the procedure,” the meaning seems to be unclear. Please identify the subject to benefit from or should be improved.

3. Line 36, not every edentulous jaw is lack of bone quality and quantity. Please rephrase the sentence.

4. Line 56, “A 57-years-old female patient” seems to be a more concise description. Line 57, “The patients’ medical chart” should be “The patient’s medical chart”. Line 66, “patients” should be “patient’s.” Line 119, “In” this way. Please consider to re-check the English format of the paper.

5. Line 233, the figure legend does not match with the figure.

6. Line 267-269, please check the English format.

7. Line 284-285, this paper did not show images of the 3-dimensional position of the inserted implants, and no comparative data was presented between the planned implants and actual position of the placed implants. Hence, there was no evidence of the “precisely inserted in depth, angulation, and crestal position.” Please consider presenting the required data to prove this point. Also, please consider presenting some information about follow-ups of the cases for supporting the safe and predictable outcomes.

Author Response

Thank you for the privilege of reviewing your work, “The workflow of a new dynamic navigation system for the insertion of dental implants in the rehabilitation of edentulous jaws: Report of two cases.” 

     This paper deals with one of the important issues in dentistry.  However, the content of this paper seems to be more suitable for Case Report rather than Research article since  (1) the workflow and the use of X-Guide had been previously described (Line 51), and (2) the result section of the abstract apparently lack of any experimental results.  Therefore, it is recommended to rewrite the paper in the structure of a case report. 

Response: The authors thank the Reviewer’s suggestion. The authors have followed the indications of case report structure usually accepted by J Clin Med (please consult the link: https://www.mdpi.com/2077-0383/8/3/398 that the authors used as a reference for structure) and in this way the manuscript benefits from a Discussion section around the advantages and disadvantages of the technique.

Other comments for minor revisions are the followings,

Line 13, The background consisted of 3 lines in one sentence, which is not easy to read. Please consider to rewrite it.

Response: The authors thank the Reviewer’s suggestion. The text was amended as suggested.

Changes: Line 14. 

Line 25, “it could benefit from improvements in order to simplify the procedure,” the meaning seems to be unclear. Please identify the subject to benefit from or should be improved.

Response: The authors thank the Reivewer’s indication. The text was amended as suggested.

Changes: Lines 23 and 24

Line 36, not every edentulous jaw is lack of bone quality and quantity. Please rephrase the sentence.

Response: The authors thank the Reivewer’s indication. The text was amended as suggested.

Changes: Line 35

Line 56, “A 57-years-old female patient” seems to be a more concise description. Line 57, “The patients’ medical chart” should be “The patient’s medical chart”. Line 66, “patients” should be “patient’s.” Line 119, “In” this way. Please consider to re-check the English format of the paper.

Response: The authors thank the Reivewer’s indication. The text was amended as suggested.

Changes: Lines 56,57 and 93.

Line 233, the figure legend does not match with the figure.

Response: The authors thank the Reivewer’s indication. The text was amended as suggested.

Changes: Line 257-258.

Line 267-269, please check the English format.

Response: The authors thank the Reivewer’s indication. The text was amended as suggested.

Changes: Lines 290,292,293.

Line 284-285, this paper did not show images of the 3-dimensional position of the inserted implants, and no comparative data was presented between the planned implants and actual position of the placed implants. Hence, there was no evidence of the “precisely inserted in depth, angulation, and crestal position.” Please consider presenting the required data to prove this point. Also, please consider presenting some information about follow-ups of the cases for supporting the safe and predictable outcomes.

Response: The authors thank the Reviewer’s indications. The information on precision of implant insertion and post-operative follow-up was introduced in the manuscript as suggested.

Changes: Lines 127-129, Figures 13, 18, 27, 31.

Reviewer 2 Report

The authors did a case report, aiming to assess the surgical workflow during maxillary full-arch rehabilitations in two patients through the All-on-4 concept assisted by DTX Studio Implant Software planning and X-Guide Navigation. The topic is clinically relevant. However, I have some comments before accepting for publication. 1. Page 3, line 69, Figure 3: please add description about the CBCT image displayed on the right side. 2. Page 4, lines 89-91: please describe the calibration process in more details, especially the sequence of calibration for each device used. 3. Pages 14-15, case 2: please consider adding post-op photos of the second case. If there are final pano / clinical pictures with final prosthesis available, please add these photos in the manuscript. 4. Discussion: please also discuss the sleeve tolerance errors from the conventional guided surgery based on the following reference: Chen et a. Accuracy of Implant Placement with a Navigation System, a Laboratory Guide, and Freehand Drilling. Int J Oral Maxillofac Implants. 2018 Nov/Dec;33(6):1213-1218.

Author Response

The authors did a case report, aiming to assess the surgical workflow during maxillary full-arch rehabilitations in two patients through the All-on-4 concept assisted by DTX Studio Implant Software planning and X-Guide Navigation. The topic is clinically relevant. However, I have some comments before accepting for publication.

Page 3, line 69, Figure 3: please add description about the CBCT image displayed on the right side.

Response: The authors thank the Reviewer’s indication. The image was not properly formatted and there is nothing to describe. The authors formatted (trimmed) the image.

Changes: figure 3 (now figure 6).

Page 4, lines 89-91: please describe the calibration process in more details, especially the sequence of calibration for each device used.

Response: The authors thank the Reviewer’s indication. The information was introduced in the manuscript as suggested.

Changes: Lines 107-112 and Figures 7-10 and 12.

Pages 14-15, case 2: please consider adding post-op photos of the second case. If there are final pano / clinical pictures with final prosthesis available, please add these photos in the manuscript.

Response: The authors thank the Reviewer’s suggestion. Actually the provided orthopantomography was the final one after one year of follow-up. The information was included in the manuscript as suggested.

Changes: Figure 31

Discussion: please also discuss the sleeve tolerance errors from the conventional guided surgery based on the following reference: Chen et a. Accuracy of Implant Placement with a Navigation System, a Laboratory Guide, and Freehand Drilling. Int J Oral Maxillofac Implants. 2018 Nov/Dec;33(6):1213-1218.

Response: The authors thank the Reviewer’s suggestion. The discussion was included in the manuscript.

Changes: 296-304